# Considerations to Model Heart Disease in Women with Preeclampsia and Cardiovascular Disease

**DOI:** 10.3390/cells10040899

**Published:** 2021-04-14

**Authors:** Clara Liu Chung Ming, Kimberly Sesperez, Eitan Ben-Sefer, David Arpon, Kristine McGrath, Lana McClements, Carmine Gentile

**Affiliations:** 1School of Biomedical Engineering/FEIT, University of Technology Sydney, Sydney, NSW 2007, Australia; clara.liuchungming@student.uts.edu.au (C.L.C.M.); Eitan.B.Ben-Sefer@student.uts.edu.au (E.B.-S.); David.arpon@gmail.com (D.A.); 2School of Life Sciences, Faculty of Science, University of Technology Sydney, Sydney, NSW 2007, Australia; kimberly.sesperez@student.uts.edu.au (K.S.); Kristine.McGrath@uts.edu.au (K.M.); lana.mcclements@uts.edu.au (L.M.); 3Sydney Medical School, The University of Sydney, Sydney, NSW 2000, Australia; 4Beth Israel Deaconess Medical Center, Harvard Medical School, Boston, MA 02115, USA

**Keywords:** preeclampsia, cardiovascular disease, heart failure, ischemic/reperfusion injury, in vivo model system, in vitro model system, ex vivo model system

## Abstract

Preeclampsia is a multifactorial cardiovascular disorder diagnosed after 20 weeks of gestation, and is the leading cause of death for both mothers and babies in pregnancy. The pathophysiology remains poorly understood due to the variability and unpredictability of disease manifestation when studied in animal models. After preeclampsia, both mothers and offspring have a higher risk of cardiovascular disease (CVD), including myocardial infarction or heart attack and heart failure (HF). Myocardial infarction is an acute myocardial damage that can be treated through reperfusion; however, this therapeutic approach leads to ischemic/reperfusion injury (IRI), often leading to HF. In this review, we compared the current in vivo, in vitro and ex vivo model systems used to study preeclampsia, IRI and HF. Future studies aiming at evaluating CVD in preeclampsia patients could benefit from novel models that better mimic the complex scenario described in this article.

## 1. Introduction

Preeclampsia is a multifactorial and dangerous disorder of pregnancy associated with increased risk of developing cardiovascular disease (CVD) in women post-partum; the risk of myocardial infarction or heart attack and heart failure (HF) is at least doubled [1]. However, its pathophysiology remains poorly understood, arising from both scarcity of patient samples from the early stages of placental development, and the variability and unpredictability of disease manifestation in existing animal models. This impedes the development of reliable monitoring and treatment strategies and limits the transferability of findings to human applications [2,3].

Currently, the main phenotypes of preeclampsia are defined as early-onset preeclampsia diagnosed before 34 weeks’ gestation, and late-onset preeclampsia diagnosed from 34 weeks’ gestation. Early-onset preeclampsia is more closely associated with abnormal placentation occurring in the early stages of pregnancy where impaired spiral uterine artery (SUA) remodeling plays a significant role, whereas late-onset preeclampsia is linked to senescence of the placenta and underlying maternal cardiovascular and metabolic disorders [4,5]. Although mechanisms of this association are still poorly understood, a recent integrative bioinformatics study identified overlapping inflammatory, angiogenesis and metabolic pathways between preeclampsia, hypertension and HF with preserved ejection fraction (HFpEF) [3]. Based on transcriptome analysis, preeclampsia phenotypes have been defined also as: (i) “maternal”, (ii) “canonical” or (iii) “immunologic” (depending on the presence of a healthy placenta and term delivery, typical features of preeclampsia or severe growth restriction and maternal antifetal rejection, respectively) [6].

Myocardial infarction leads to myocardial tissue damage, with loss of cardiomyocytes [7]. Ischemic cardiomyopathy is the most common cause of heart failure and occurs when blood flow to the myocardium is decreased or blocked to a section of the heart [8]. The effective therapeutic intervention is immediate myocardial reperfusion such as percutaneous coronary intervention (PCI) by restoring the blood flow. However, this process can increase the oxygen level in the heart at a toxic rate, leading to ischemia-reperfusion injury (IRI), and induce further cardiomyocyte death [9,10]. The current treatments are successful in reducing immediate mortality, but there is no effective therapy preventing myocardial reperfusion injury, including subsequent scarring and the necrosis of the heart muscle leading to chronic HF.

HF is characterized by the irreversible damage to the ventricular muscle wall [11]. This is often hallmarked by changes in heart shape and size, cardiac remodeling, increased ventricular myocardial mass, hypertrophy, increased collagenous scar tissue and fibrosis [12,13]. HF is considered a chronic phase of cardiac impairment, secondary to other CVDs, rapidly growing in both confirmed and suspected undiagnosed cases, including in the presence of and post preeclampsia [14,15]. At present, prevention and treatment of the underlying CVD factors remain the only way to treat HF, while its prevalence is estimated to have doubled from 27 million cases worldwide to over 50 million cases, with a one in five lifetime risk of developing HF [11,15,16]. Over the past few decades, a better understanding of IRI and HF pathogenesis has been made possible due to several representative in vitro and in vivo models. However, these models have limitations for treating IRI and HF [17,18,19,20,21], which will be discussed below.

In pregnant women, the causes of preeclampsia-induced cardiac dysfunction are still unknown, mainly due to the lack of optimal models to recapitulate this complex disease in the laboratory. Nevertheless, what has been identified is that systemic oxidative stress, inflammation and irregular angiogenesis present in preeclampsia can lead to cardiac fibrosis, apoptosis, diastolic and systolic dysfunction, and subsequent HF post-partum [4,5]. The mechanisms of these aberrant processes that lead to HF following preeclampsia require more reliable disease models to further explore the mechanisms of onset and progression as well as provide an advanced platform for treatment development [22,23].

Advancements in the field of tissue engineering have elevated in vitro models to the point that they are now a promising alternative to in vivo models—that is, animal experimentation [24,25,26]. The choice of the most appropriate methodology will depend on multiple factors including the specific research question, availability of equipment and skills, budget and time restrictions. This review will primarily compare and contrast the existing in vivo and in vitro experimental models for preeclampsia, IRI and HF, outlining their strengths and limitations. Subsequently, the development of future models linking these disease entities together will be discussed.

## 2. Preeclampsia Models

### 2.1. In Vivo Models

The most reliable animal model of preeclampsia should closely mimic the pathogenesis of the disease and the clinical signs and symptoms. Preeclampsia is currently difficult to study due to its multifactorial nature and a lack of suitable patient samples because taking placental samples during pregnancy is invasive and can increase the risk of miscarriage. The commonly used laboratory species do not develop spontaneous preeclampsia. Mouse models display variable and unpredictable disease manifestation, with limited transferability to human applications. Currently, there are numerous animal models that have been developed to simulate these characteristics, however all representing different features of preeclampsia.

#### 2.1.1. Animal Trophoblast Invasion Model

The unique process of placentation in humans makes it challenging to establish a reliable animal model recapitulating human trophoblast invasion. Inappropriate placental development is a fundamental feature in the preeclampsia pathogenesis, where the role of the trophoblast cells in remodeling the maternal uterine vasculature to support growth and development of the fetus is impaired [4]. Some species possess hemochorial placentation where the trophoblasts infiltrate the uterus and create intimate connections with the maternal vasculature, or endotheliochorial placentation, which is less invasive [3,27,28]. This is commonly seen in mammals including rodents, primates, insectivore and bats. Other species possess epitheliochorial placentation where there is a separation between the trophoblastic tissue and maternal tissue in ruminants, pigs and other domesticated animals [3,28,29].

Rodents are good models of preeclampsia as they display interstitial and endovascular trophoblast invasion leading to maternal artery remodeling, and also possess hemochorial placental types similar to humans [3,29]. These models have enabled a better understanding of the pathophysiology of the disease, however all of these models are induced and are not able to recapitulate different phenotypes of preeclampsia. Preeclampsia can be induced in rodents surgically, environmentally, genetically or immunologically [30].

#### 2.1.2. Utero-Placental Ischemia Model

Utero-placental ischemia is another key feature in the pathogenesis of preeclampsia, particularly early onset preeclampsia, leading to hallmark complications including high blood pressure, vasoconstriction, and endothelial dysfunction [31]. Early models of this kind mainly focus on abruptio placentae in animals such as baboons, rabbits, rhesus monkeys, and dogs. This type of model of preeclampsia is induced by either a temporary or permanent ligation of uterine arteries and/or aorta to induce high blood pressure and proteinuria during pregnancy [32,33,34,35]. The method has been modernized many times over the years to the most utilized method now being the reduced uterine perfusion pressure (RUPP) model developed by Granger et al., [28,36]. Based on the murine model developed by DJ Eder and MT McDonald [37], the method was modified to assess how hypoxic conditions correlate to cardiovascular and renal dysfunction [28,37]. RUPP models have been used to study and evaluate hallmark features of preeclampsia including elevated blood pressure, proteinuria, fetal growth restrictions, intrauterine growth restrictions, histopathological placental aberrations, reduced placental and embryo weight, and glomerular endotheliosis [28].

#### 2.1.3. Anti-Angiogenic Response Model

One of the key underlying processes in preeclampsia include angiogenic imbalance. The most well studied angiogenic factors, vascular endothelial growth factor (VEGF) and pregnancy induced growth factor (PIGF), are often decreased, whilst anti-angiogenic factors such as soluble fms-like tyrosine kinases (sFlt-1) and soluble endoglin (sEng) increased. Novel angiogenesis-related pathways implicated in preeclampsia have also emerged, including the FKBPL-CD44 pathway [38]. This angiogenic imbalance where anti-angiogenic proteins are increased and pro-angiogenic proteins are reduced often leads to endothelial dysfunction [39].

VEGF is responsible for the production of nitric oxide and other vasodilatory molecules, which are key in maintaining low vascular tone and blood pressure, hence facilitating appropriate glomerular function [38,39,40]. To study this, models have been developed that focus on administering exogenous anti-angiogenic factors (such as sFlt-1 and sEng) in pregnant rats to induce a preeclampsia-like phenotype [39,40,41,42]. Administration of sFlt-1 adenovirus in pregnant rats is capable of generating the preeclampsia phenotype, including decreased levels of VEGF and PIGF, with sFlt-1 having a direct impact on maternal endothelium [39,40]. Similarly, sEng is associated with a reduction in endothelial nitric oxide synthase (eNOS) activity and vascular tone that can lead to vascular damage and dysfunction [40,42]. These models are best utilized for studying the downstream effects, pathophysiology and treatment options for preeclampsia as opposed to its underlying causes [3].

#### 2.1.4. Immune Models

Several models investigating the immune response in preeclampsia pathogenesis have also been developed. Both TNF-α and IL-6 are inflammatory cytokines elevated in the presence of preeclampsia, a feature recapitulated in animals’ models that also exhibit the typical preeclamptic features, including elevated blood pressure, proteinuria, and elevated sFlt-1 and sEng levels [43,44,45,46,47]. Conversely, IL-10 knockout mice in hypoxic conditions in pregnancy displayed preeclamptic symptoms, yet fetal growth restrictions were only seen in wild-type mice exposed to hypoxic conditions [48].

Some women with preeclampsia also display autoantibodies to phospholipids and angiotensin II type I receptors increasing the disease risk. Immunization against these antigens has been used to create other types of immune models of preeclampsia [49,50]. These types of models have suggested a pathophysiological association between immune factors and hypoxic conditions [50]. Hypoxia has been shown to decrease the number of trophoblast cells by causing cell death and as a result of a maternal allogenic immune response induced by shedding of paternal antigens from the placenta into maternal circulation [51].

### 2.2. In Vitro Models

As described above, one of the key biological processes in pregnancy includes placental development; irregular placental development and growth have been closely associated with preeclampsia [52]. A unique feature in human placentation is endovascular invasion, where trophoblast cells invade the decidua and myometrium to remodel the spiral uterine arteries. Trophoblast invasion is a complex process involving interactions with numerous different cells including endothelial, immune, stem and other stromal cells in the body, making it challenging to study. Furthermore, there is high variability in placentation between species, hence a lack of good animal models to recapitulate early human trophoblast invasion and development [3,52]. Due to these limitations, a wide range of in vitro models have been developed using trophoblast, endothelial, immune and recently mesenchymal stem cells to study mechanisms leading to inappropriate placentation that could lead to preeclampsia [4,23]. These models of placental development and growth have been utilized to investigate the mechanisms of development of different trophoblast cell lineages, invasion and differentiation of trophoblast cells, the formation of syncytia, morphogenesis, placental development, endocrine function, maternal immune response, metabolism and transport, and disease adaptation. Important diagnostic biomarkers and/or potential therapeutic agents have also been identified and developed using these models [28].

#### 2.2.1. Models of Trophoblast Cells

Many in vitro models used to study preeclampsia are based on using trophoblast cells [52,53,54,55,56]. These cells can be obtained through a culture of primary villous and extravillous trophoblasts as well as trophoblast cell lines obtained from the placenta. Fresh isolation of primary trophoblasts is challenging and difficult to obtain, and is often representative of term placental trophoblasts rather than first trimester trophoblasts important for SUA remodeling and appropriate placental developments. Trophoblasts are dynamic cells, which undergo many rounds of differentiation and interact with various cells at different gestational points during placental development. Several freshly isolated first trimester trophoblasts have been transfected with the virus, acquiring the advantage of longer proliferation in cell culture than primary cells [4].

There is controversy around the types of cell lines that should be used to study the role of trophoblasts in SUA remodeling and placental development. The optimal in vitro model would utilize freshly isolated first trimester (primary cytotrophoblast) trophoblast cells; however, when primary cells are newly isolated and cultured, they fuse impulsively, forming syncytiotrophoblasts [53]. Further limitations include their inability to divide, their limited lifespan, and ethical considerations in the difficulty in obtaining human placental tissue to isolate these cells, especially early in pregnancy [54]. The vast majority of trophoblast cells are isolated from the term placentae following delivery of the baby, which is not reflective of their role in placental development. Inadequate function of trophoblasts in remodeling SUA and placental development is one of the key processes leading to preeclampsia.

The many different cell lines used in placental research include choriocarcinoma (trophoblastic cancer of the placenta) cell lines (JEG-3, BeWo, and JAR), utilized for their villous and extravillous trophoblast features (VTs and EVTs). However, the gene expression profile of choriocarcinoma cell lines inadequately reflects VTs and EVTs due to their inconsistencies in the transcriptome profile, malignant behavior, high passage via hamster cheek pouch, and atypical chromosome count [52,55]. Still, choriocarcinoma cells prove useful to study particular facets of trophoblast immunobiology. JAR cells were found to be beneficial in the investigations of the fusion of the syncytial VT layer and JEG-3 have been beneficial in identifying EVT HLA class-1 molecules [55,57].

HTR-8/SVneo cell lines contain a combination of EVTs transfected with a retrovirus plasmid (simian virus 40) aimed at acquiring the advantage of longer proliferation in cell culture (56), which are frequently used to study EVT invasion, proliferation, and regulation. HTR-8/SVneo is now considered outdated in its attributes to EVTs [57]. A more appropriate trophoblast cell line appears to be the ACH-3P cell line, which was developed by fusing freshly isolated primary first trimester cells with a human choriocarcinoma cell line (AC1-1) [58]. These cells have been shown to closely mimic primary trophoblasts, express trophoblast markers including cytokeratin-7, integrins and matrix metalloproteinases, and display appropriate invasion potential and primary trophoblast transcriptome profile. Interestingly, this cell line contains both VT and EVTs, which can be separated by the presence of HLA-G on the cell surface. When a range of cells were compared to healthy term and first trimester placentae, chromosome 19 miRNA cluster (C19MC) and C14MC that correlate with gestational age were expressed accordingly in HTR-8/SVneo and ACH-3Ps and were absent from choriocarcinoma cells, questioning their reliability for use in trophoblast studies [59]. Furthermore, when functional aspects of choriocarcinoma cells (BeWo, JAR JEG-3) and ACH-3Ps were compared, BeWo cells were determined as the most suitable model of syncytial fusion, whereas ACH-3P and JEG-3 were representative of primary cells in terms of barrier function; overall, ACH-3Ps were deemed the most reliable for placental nanoparticle transport studies [60].

#### 2.2.2. Placental Explants

The use of placental explants involves obtaining a small section of placental tissue to be cultured ex vivo in a dish. This method is used to study trophoblast proliferation and invasion where the cells are maintained in their adequate cellular environment. In the past, placental explants have been used to study placental functions and mechanisms including cellular uptake and interactions, disease mechanisms through secretome profiling and genetic manipulations, as well as drug effects and toxicity. The most common use of this technique nowadays is in relation to trophoblast function. The main advantages of placental explant models are that the trophoblasts are conditioned in a co-cultured environment with appropriate cells, enabling investigations of function and behavior. The main limitations of this model include an inability to separate functions, mechanisms and responses of individual cell types given that placental tissue is multicellular. Furthermore, the explants are generally obtained from term pregnancies, preventing identification of processes implicated in early placental development, which are closely associated with preeclampsia. Extensive degradation of the cells is also observed within as little as 4 h and cell death can occur within 48 h [53,61].

#### 2.2.3. Microfluidics Models

Microfluidic-based assays recapitulate cell–cell and cell–stroma interactions and can monitor cellular and molecular changes in real-time. This facilitates profiling of the secretome in real-time and not just at one point in time. These models allow for the control and manipulation of physical and chemical factors, which are involved in processes at the microenvironment level [52]. The main advantages are the ability to generate chemical gradient profiles, identify chemotactic factors associated with different cell types and observe individual cell migration and morphology while utilizing a cost-effective method [62]. In relation to the study of preeclampsia, this model captures critical features of the maternal–fetal interface, structural placental characteristics, and some physiological features [63]. The device is limited by the amount of stress that can be induced in this model in comparison to what is endured during pregnancy and preeclampsia. It is limited in size to replicate the shear force that is observed in fetal capillaries under physiological conditions [63]. A microfluidics model of placental vasculature and growth incorporating three different cell types (fibroblasts, endothelial cells and pericytes) was recently developed, capable of demonstrating the inflammation-mediated vascular leakage and leukocyte infiltration of the placenta, processes associated with preeclampsia [64]. Another microfluidics model more closely resembling the placenta was developed to include human choriocarcinoma trophoblast cells (JEG-3) and human umbilical cord endothelial cells (HUVECs) seeded between extracellular matrix membrane under dynamic flow conditions illustrating epithelial and endothelial layers within the placenta [63].

#### 2.2.4. In Vitro Models of Endothelial Dysfunction in Preeclampsia

Endothelial dysfunction is another key underlying cause of preeclampsia induced by increased levels of antiangiogenic factors, sFlt-1 and sEng, and a reduction in angiogenic factors, PIGF and VEGF [65]. Another likely cause of endothelial dysfunction includes placental hypoxia or ischemia-reperfusion, leading to oxidative stress, inflammation and endothelial dysfunction [66]. Endothelin 1 (ET-1) and vascular cell adhesion molecule 1 (VCAM-1) are markers of endothelial dysfunction that are also elevated in preeclampsia. ET-1 is a powerful vasoconstrictor released from endothelial cells and VCAM-1 is a cell surface adhesion molecule involved in leukocyte-endothelial cell signal transduction [67]. Both of these molecules, when elevated in preeclampsia, can lead to hypertension and a reduction in blood flow to the major organs. If left untreated, end organ damage likely follows. Research investigating therapeutic options for endothelial dysfunction utilizes either HUVECs or uterine microvascular endothelial cells treated with combination therapy to help restore angiogenic balance and hence ameliorate endothelial dysfunction in preeclampsia [65,67]. Brownfoot et al. [68] previously identified metformin and sulfasalazine as each individually reducing secretion levels of endothelial sFlt-1 and sEng [69]. Recently, they utilized a combination therapy of metformin and sulfasalazine, demonstrating a reduction in sFlt-1 and overexpression of VEGF-α from the placenta. Individual low dose administration led to a reduction in sEng and an increase in PIGF; however, no additive effect was observed from the combination therapy. Moderate effects were noted on reducing markers of endothelial dysfunction with some reduction in ET-1 observed and no change to VCAM-1. The researchers suggest that this low dose treatment was potentially too low and higher doses may be required for the desired effects [65]. In a similar study, proton pump inhibitors were shown to be able to ameliorate TNF-α-induced endothelial dysfunction of HUVECs and uterine microvascular cells, by blocking VCAM-1 expression, leukocytes adhesion to endothelium and irregular tube formation [70]. Other pregnancy-safe medicines including pravastatin were also investigated for the same purpose and showed promising results in restoring endothelial functional dysfunctions [67,71,72].

Co-culture models with trophoblast and endothelial cells are also utilized to mimic the interaction between these two cell types during SUA remodeling that leads to the replacement of maternal endothelial cells to establish high-caliber, low-resistance vessels that enable increased blood flow to the developing feto-placental interface. In this context, a recent study showed the importance of integrins α1β1 in trophoblast and endothelial cell interaction. Human uterine myometrial endothelial and trophoblast cells (HTR-8/SVneo) were labelled with different fluorescent stains; HTR-8/SVneo were pre-treated with various neutralizing integrin antibodies before co-culturing with endothelial cell networks formed in Matrigel. Trophoblast integration into the endothelial cell network was assessed and the expression of various invasive pathways was determined, identifying galectin-1, TIMP-1, PAI-1, MMP-2, and MMP-9 as key in this process [73].

Despite the great utility of in vitro models, the multifactorial nature of the disease involving many different organs in addition to the placenta limits their use in recapitulating all features of preeclampsia. For this reason, the development of reliable in vivo models is also important [3].

### 2.3. Additional Considerations of Current In Vivo and In Vitro Models

In summary, abnormalities of cellular and molecular origins in preeclampsia occur between weeks 8 and 18 of pregnancy. It is very difficult and rare to obtain samples of placental tissue during this early stage of pregnancy, which is a major obstacle in the study of the disease [74]. Most methods rely upon placental specimens obtained after delivery. This is limiting in terms of the knowledge that can be obtained within this field, often representative of consequences of preeclampsia rather than pathogenesis. Inherent complications with in vitro and in vivo models require the development of novel model systems that are low-risk, low-cost and reproducible [28]. Since there is no model that can comprehensively replicate the complexities of the disease, a range of in vivo and in vitro systems and preeclampsia-like models are currently necessary to be used in parallel to elucidate the pathophysiology of preeclampsia [3].

## 3. Cardiovascular Models to Mimic Ischemic-Reperfusion Injury

### 3.1. In Vivo Models of Ischaemic Heart Disease

While animal models have been extensively used to assess various parameters of cardiac cell physiology and electrophysiology within a living organism, as they integrate the complexity of the whole organism and allow long-term studies, they also display several limitations (Table 1). In addition to ethical considerations, these models do not fully emulate human physiology, are expensive and need experienced personnel [17,75].

Myocardial IRI is induced in animals by using a suture to temporarily occlude the left descending coronary artery for the designated ischemic time, which is then subsequently released to allow reperfusion [75,76]. This approach captures the process of hypoxia-reoxygenation typical of IRI. However, it does not fully recapitulate the clinical setting, perhaps due to the vessel stenosis or occlusion of an atherosclerotic artery by dislodged plaque, and the reperfusion by the PCI in humans. The ischemic time, ischemic preconditioning and the duration of the reperfusion may depend on the species and other factors, including gender, age and temperature [77,81]. For example, the ischemic time for animal models spans between 30–40 min (mice, rats and rabbits) and 60–180 min (dogs, pigs and monkeys), whereas in humans it is around 198–411 min [81]. Moreover, animal hearts have physiologically different hearts that can affect their response to IRI [75].

#### 3.1.1. Small Mammals Models

Small mammalian animal models, including mice, rats, hamsters and rabbits, have been extensively used to identify effective therapeutic interventions to regenerate the heart following injury [17,82]. Mice are frequently utilized for IRI experiments, as they are genetically malleable, have a rapid breeding cycle and are cheaper compared to other bigger animals [83]. However, they fail to fully replicate human pathophysiology and morphology [8,21]. For example, rodents and mice have a higher heart rate, and different cardiac basal metabolism and electrophysiology compared to humans [75,83,84]. Furthermore, myocardial ischemia develops faster in rodents and is completed after 30 min of coronary occlusion [81]. Using small mammals for IRI allows the examination of the interactions of various cell types, testing of drug effects in the whole organism by analyzing the cell biological and molecular mechanisms as well as genetic modification of the animals. Despite this, the implications for drug efficacy and safety in humans remain to be determined, and while large animal models are considered closer to the clinical settings of IRI in humans, smaller animals are preferred for early testing of feasibility and safety [8,19].

#### 3.1.2. Large Non-Human Mammals Models

Large animals including dogs, sheep, pigs or primates have been used for testing preclinical therapeutic approaches [8,19]. Pigs are the closest analogues to humans and have gained an increasing relevance in recent years as models of IRI since they have a comparable heart size, heart rate, and do not present with resistance against infarction that is typical of primates [75,81,85]. Studies using dogs, sheep, and pigs focusing on the evaluation of novel stem cell therapy approaches to treat ischemic heart disease have been shown to be relevant to humans with a better translation into the clinic [84]. Despite this, preclinical large animal models fail at mimicking remote ischemic conditioning and the inflammatory response [85]. Furthermore, patients with IRI and frequently observed co-morbidities, including diabetes, hypertension and renal failure, are routinely treated with other drugs, hence masking some of the effects of IRI in research animals [84].

### 3.2. Ex Vivo Models of Ischemic Heart Disease

A Langendorff preparation used to mimic IRI ex vivo involves the isolation of the whole heart from an animal and its perfusion to simulate blood flow [82]. This model allows the evaluation of cardiac function for a more physiological simulation of IRI and for the study of the effects of several drugs to protect against IRI [82,86]. The benefits of this model are that it is low cost, simple to prepare, reproducible and can examine the heart in isolation from the other organ systems and independently of the exocrine control. However, the absence of the reduction-oxidation (redox) signaling and other paracrine factors besides the use of animal cells limits the translation of these studies into humans [17,86]. Additionally, a Langendorff preparation might be viable for only several hours, and around 5–10% of deterioration in the chronotropic and contractile function is developed per hour [86,87].

### 3.3. In Vitro Models of Ischaemic Heart Disease

In vitro models of myocardial IRI are essential to study the direct effect of reperfusion on individual cardiac cells including cardiomyocytes and to identify potential novel therapeutic targets to prevent the subsequent irreversible cardiac damage [76]. These can be used to control and identify individual external factors that may be involved in IRI and are commonly divided into two-dimensional (2D) and three-dimensional (3D) cell-culture models, as shown in Table 2.

#### 3.3.1. Cardiomyocytes Cell Culture (2D Culture)

The 2D in vitro models based on cell monolayers using either freshly isolated primary cardiomyocytes or cell lines in a non-physiological setting induce ischemia with hypoxic conditions and reperfusion with reoxygenation [8]. While freshly isolated cardiomyocytes are more relevant to cardiac cell lines, repeated experimental observations cannot be carried out using the same primary cells [93]. The models primarily rely on isolated cardiomyocytes from animals, immortalized cell lines, or human induced-pluripotent stem cells (hiPSCs). The advantage of studying isolated cardiomyocytes is that it allows precise control of the cellular and extracellular conditions, without the influences of other cell types, notably endothelial cells, fibroblasts, inflammatory/immune cells and platelets as well as circulating factors including hormones, cytokines and neurotransmission [8,88,89,93]. Two-dimensional cardiomyocyte cell culture models offer the advantage to identify the effects of therapeutic agents on cardiomyocytes to elucidate molecular signaling pathways, assess drug-induced cardiotoxicity and achieve targeted manipulation of gene expression involved in IRI in order to determine disease mechanisms [19]. However, isolated cardiomyocytes in in vitro 2D cultures are removed from their surroundings, that is, syncytial neighbors, blood vessels and extracellular matrix. This leads to the loss of important cues for the optimal pathophysiological scenario typical of IRI [93,94]. Therefore, findings of studies conducted at a cellular level may not necessarily predict what the response would be if the same drug, toxin or signaling agent is tested in in vivo models. Nevertheless, the most attractive cell source for IRI studies are cardiomyocytes derived from hiPSCs (hiPSC-CMs) as they can be cultured to generate human IRI model system, with potential for personalized medicine and be patient-specific [76,95]. This personalized model could be applied for prediction and drug screening of women’s heart disease in high-risk women post-preeclampsia given that there is a substantial inter-patient variability for the future risk of CVD.

#### 3.3.2. Three-Dimensional (3D) Cultures

Tissue engineering employed for the purpose of investigating cardiac modelling facilitates the generation of 3D structures from cardiomyocytes alone or in co-cultures with other cell types including endothelial cells and fibroblasts [89,96]. Cardiac cells can be grown in scaffolds, scaffold-free or matrix environments aiming to mimic the extracellular matrix (ECM) aspects of the heart. For example, scaffolds made of collagen and fibrin provide a 3D environment for cells to attach onto, interact with other cells and conduct electrical signals [25].

Engineered heart tissues (EHT) comprised of cardiomyocytes embedded within fibrin-based constructs have been used in numerous in vitro and in vivo studies [90,97,98,99,100]. In particular, EHTs have been used to study features typical of IRI and is a promising model to study cardiac function in vitro [90]. The advantages of using EHT are that they are easy to fabricate and can provide reproducible results within a short timeframe [90]. EHTs can be used for real-time measurements of the contractile function and may represent a promising tool for advancing the treatment and prevention of IRI [76].

Scaffold-free approaches using spheroids in hanging drop cultures provide similar advantages to EHTs and do not require the addition of foreign material for the fabrication of the cardiac tissue [88,101]. For instance, Jeong et al.’s [102] research showed that genetically engineered antigen-1-positive cardiac stem cells (Sca-1^+^ CSC lines) secrete paracrine factors such as SDF-1α and have cardioprotective roles described using in vitro spheroids. SDF-1α has demonstrated to protect the ischemic cardiomyocytes by inducing the signaling pathway for cell growth, survival, and protein synthesis in the 3D IRI spheroid heart model [102].

Interestingly, hiPS-CMs retain a fetal phenotype and are more resistant to IRI hence limiting the translation of findings using these cells into humans [76,95,103]. Tissue engineering approaches using bioreactors or microfluidics devices allow further maturation of hiPS-CMs into a more adult phenotype, but more progress needs to be made as they cannot recapitulate complex features typical of the in vivo microenvironment, such as adult cardiomyocyte function [10,91]. Bioreactors or microfluidic “organ-on-chips” devices provide precise control and better recapitulate the cellular microenvironment of IRI of the heart compared to other in vitro systems, including the monitoring of critical parameters such as pH and oxygen levels [76,104]. However, these cultures do not fully represent the complexity and tissue architecture of the heart due to the absence of different cell types such as fibroblast, endothelial cells and immune cells [91]. Co-culturing cardiomyocytes with other cell types found in the human heart improves the representation of the key aspects of the phenotypical and cellular heterogeneity as well as microenvironmental cues, hence leading to a more reliable model of IRI in the human heart [104]. Therefore, future studies aiming at better engineering strategies of the human heart microenvironment are needed to improve translation of the findings using IRI in vitro models into patients.

### 3.4. Additional Considerations of Current In Vitro and In Vivo Models

A common criticism of preclinical IRI studies (both cell culture and animal models) is that testing is performed using a homogenous, young and healthy population, while myocardial infarction primarily affects a diverse older population with different comorbidities, including diabetes and cardiac hypertrophy [76]. These additional factors affect the cardiomyocyte response, including their susceptibility to reperfusion injury and the effectiveness of treatments, and further considerations are needed to better translate preclinical studies into humans [95,103,104].

## 4. Cardiovascular Models to Mimic Heart Failure

Although it is not clear whether endothelial dysfunction is a cause or consequence of preeclampsia, longitudinal studies reported that endothelial dysfunction can persist for 10–20 years following preeclampsia in pregnancy [105]. Epidemiological evidence shows a strong association between preeclampsia and future CVD including HF; however, the mechanisms are poorly understood. In a recent bioinformatics study based on publicly available datasets, 76 overlapping biomarkers which translated into 29 shared pathogenic pathways were identified between preeclampsia, hypertension and heart failure with preserved ejection fraction (HFpEF) [105]. Nevertheless, there are limited studies that investigate the pathogenic mechanisms between preeclampsia and HF, and this area of research is also lacking reliable in vitro and in vivo models.

### 4.1. In Vivo Models

Animal models seek to mimic both the pathological features of HF and the clinical scenario of patients with HF. The use of animals for this type of modelling can range from multi-organ level (e.g., nervous input to the cardiovascular system) to cellular level (e.g., variable expression of cell-specific genes) [106]. In vivo models exhibit a range of unique advantages including physiological relevance in large animals and reliable standardized protocols in small animals, features that in vitro models are still tackling [20]. Small animal models typically utilize either mice or rats to perform a surgical procedure and induce HF [107]. One of the most frequently used methods is the transverse aortic constriction surgical procedure (TAC). Originally developed by Rockman et al. [108], this widely used method reliably induces HF via increased left-ventricular (LV) afterload, resulting in a sharp increase in LV mass as early as two weeks [108,109]. The disadvantages of these models include the inability to induce progressive features of HF, and they are therefore mainly used for testing the role of specific proteins involved in cardiac dysfunction and genetic mechanisms via transgenic mice [110,111]. Other less common surgical procedures in rats, including a left coronary ligation (LCL) and ascending aortic constriction (AAC), offer a relatively simple procedure with low cost, allowing for a greater number of subjects without damaging large volumes of myocardial tissue [112,113]. LCL induces HF via MI while AAC induces HF via a pressure overload in the ventricles; unlike TAC, this pressure is gradual as opposed to acute. However, these models typically require expensive equipment for analyses, and results are less likely to be reproduced in the clinic. This also posits the challenge of inducing only a specific phenotype of HF, which may be characterized by either a reduced (HFrEF) or HFpEF.

The suitability of large animal models is dependent on the research question being addressed. Spannbauer et al. [114] postulated that the use of large animal models in studies requiring genetic manipulations is limited due to their long gestational periods. Additionally, the development of transgenic species requires highly skilled researchers and the use of expensive specialized facilities that may be a limitation from a resource perspective. However, large animal models have a better translational potential to human clinical studies [20]. Furthermore, these models are considered optimal for studies requiring implantation of sensors and data gathering in a non-anesthetized state related to long term outcomes and medical device development [18,115]. Similar to small animal models, aortic constructions resulting in LV remodeling and hypertrophy are commonly utilized to induce HF, though there are other methods not utilized in mice and rats [106,115]. Pacing-induced tachycardia has been recognized as a method of HF modelling in large animals via the onset of dilated cardiomyopathy [116]. This model reliably results in progressive and reversible human HF hallmarks such as mechanical, structural and hormonal alterations that has previously been used for testing pharmacological therapies, though it does not result in myocyte hypertrophy or fibrosis and is reversible unlike human HF [117,118].

### 4.2. In Vitro Models

Modelling HF in vitro has been particularly challenging for researchers due to the nature of the disease and limited options compared to IRI models. As HF is largely related to cardiac output, current methodologies to model and study this aspect are limited. The advent of iPSC-derived cardiac myocytes has propelled in vitro models forward via a reliable source of human cardiac myocytes [26].

Two-dimensional cultures have typically fewer representative models of clinical outcomes but are cost-effective, with access to high throughput assays. This is largely due to the availability of transformed cell lines that, although they may have lower resemblance to in vivo counterparts, allow for unrestricted proliferation. For this reason, transformed cells are often used for drug discovery and cardiotoxicity studies, while iPSC-derived cells are used for genetic and functional studies [25,119]. This is achieved by treating cells with ET-1 to induce cardiac hypertrophy, one of the major risk factors for HF [120].

While useful for investigating certain biochemical or genetic changes, 2D cell cultures ultimately lack important physiological aspects including cell–cell and cell-ECM interactions [102]. An emerging trend that has been rapidly gaining recognition for its potential to model diseases that addresses the drawbacks of 2D cultures is the use of 3D cell cultures of spheroids. Cardiac myocytes cultured within 3D environment often employ a biomaterial such as a hydrogel or biocompatible polymer to mimic the ECM, providing a 3D architecture for cell spheroids to interact in all spatial dimensions both with other cells and their environment. Various ECM are employed for fine-tuning of the microenvironment by modifying properties including elasticity, stiffness, conductivity and porosity [25]. This is a promising advancement for HF modelling as reports have demonstrated the utility of 3D cell models capable of simulating blood flow with modified ejection fractions, observing contractile forces and relaxation velocity in cardiac myocytes as well as variable mechanical cues to simulate increased afterload [121,122,123].

With the increase in controllable parameters, there is also an increase in complexity. Lack of standardized protocols compared to 2D cell models means that experimental design is more demanding and without high-throughput testing. Additionally, direct induction of HF is still a challenge for disease modelling. The primary methods of modelling in vivo HF rely on replicating the disease by primarily using either cardiovascular high-risk factors including hypertension and MI. Additionally, HF can be induced by mimicking its phenotypic presentation, such as reduced ejection fraction (to simulate the heart as a failing blood pump). Though useful strategies, these approaches restrict studying complex presentations such as HFpEF and long-term disease state.

Overall, in vitro models have great utility as they provide a platform for fundamental biology in drug development, biochemical, genetic and pathophysiology studies, free of ethical concerns for animal use with the potential to reduce the number of animals used in these experiments. Challenges remain in the generation of high-throughput methods for 3D cell analysis and maturation of iPSC-derived cardiac myocytes beyond the neonatal phenotype that is commonly observed.

### 4.3. Additional Considerations of Current In Vivo and In Vitro Models

In vivo models provide information regarding HF pathophysiology and progression that cannot be replaced until in vitro technologies are developed further. Although there are ethical concerns with animal use, relatively cheap small animal models are still the primary source of data for studying signal cascades and biological processes, while large animals are paramount for studying contractile function that cannot be replicated in small animals due to biological differences in functional activity. Further information on the advantages and limitations of various in vitro and in vivo models of HF are presented in Table 3.

## 5. Discussion

As discussed above, different types of models currently used for preeclampsia, IRI and HF have provided knowledge in understanding some of the pathophysiology and the mechanisms of these CVDs and have been utilized extensively for drug screening. However, none of the current models could fully replicate all the pathophysiological mechanisms for human applications. Following such an extensive list of models in preeclampsia, IRI and GF, we could not identify a model that fully recapitulates the link between preeclampsia and future CVD in laboratory models, which is critical to further address the heterogeneity typical of the preeclampsia phenotypes described in the Introduction. A wide range of in vitro, ex vivo and in vivo model systems of individual diseases have helped in the discovery and validation of novel therapeutic targets and disease mechanisms. However, there are still large gaps in the knowledge or technology impeding successful translation of the findings from preclinical into human studies. While the etiology of preeclampsia is still not fully uncovered, there is evidence that the fetal environment plays a major role [126]. Among the plethora of potential biomarkers identified in previous studies, future studies will be critical for the identification of better markers for improved classification and potential prevention and therapeutic approaches [127,128].

The approach used to mimic IRI in in vitro and in vivo model systems does not fully recapitulate the occlusion and the reopening of the blood vessels. Furthermore, most tests are performed using a homogenous, young and healthy population without the inclusion of comorbidities. In terms of HF, the current model systems are unable to induce progressive features of HF. The advent of iPSCs could generate in vitro models with potential for personalized medicine [76,95]. Preeclampsia is more complex to replicate in a model, because it starts to develop between 8 and 18 weeks of pregnancy and sampling of the placental tissue is challenging during this stage [74]. There are no current models that mimic the complexities and pathophysiology of the disease.

In vitro models are useful for investigating biochemical or genetic changes, but 2D cultures lack important physiological aspects including cell–cell and cell-ECM interactions [101]. The advancement of 3D cultures has propelled in vitro models forward and could potentially fully replicate the microenvironment and physiology of human heart however further improvements in research are still needed. The most efficient models remain in vivo animal models, especially with large animals that have translation potential towards human clinical studies. However, the reliability of these models still remains unclear, as they do not fully mimic the pathogenesis of the disease, clinical signs, and symptoms.

The pathophysiology of preeclampsia still remains poorly understood, impeding the development of much needed monitoring and treatment strategies for human applications [2,3]. Moreover, preeclampsia is associated with higher risk of subsequent hypertension, MI and HF, leading causes of death in women [129]. The mechanisms of this association are poorly understood due to the lack of available models of disease. Hence, it is critical that further research is carried out that can lead to the development of new platforms and to shed a better light on the pathophysiology leading to the subsequent development of CVD post preeclampsia.

## 6. Conclusions

The existence of multiple disease models for preeclampsia, IRI and HF illustrates the inherent complexity of these diseases. As discussed above, none of these models encompass all the key pathophysiological mechanisms. However, our increasing knowledge, the diversity of techniques and approaches of each pathology would allow us to develop a unique model that could demonstrate the link between preeclampsia and the subsequent development of CVD. Three-dimensional personalized models based on patients’ own cells and iPCSs could better recapitulate the complex cardiovascular scenario typical of preeclampsia and be developed further for testing novel therapies, which is particularly challenging in human pregnancy (Figure 1).

## Figures and Tables

**Figure 1 cells-10-00899-f001:**
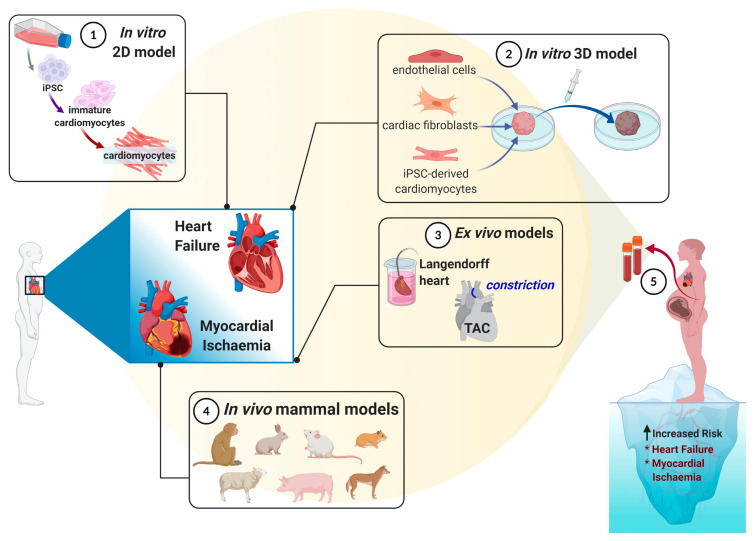
Modeling cardiovascular complications in preeclampsia women. Despite the several in vivo, ex vivo and in vitro models to mimic cardiovascular complications, future studies utilizing personalized approaches, such as patient-derived cells, may benefit to further advance the development of novel therapeutics to both prevent and treat preeclampsia-associated cardiovascular disease in women.

**Table 1 cells-10-00899-t001:** Major advantages and disadvantages of in vivo models of ischemic heart diseases.

Model	Typical Features	Advantages	Disadvantages	References
Large Nonhuman mammals (dogs, sheep, pigs or nonhuman primates)	-They capture the process of hypoxia-reoxygenation but does not fully model the clinical setting. - Interactions between various cell types.	-Pigs are the closest analogues to humans, followed by sheeps and dogs (comparable heart size and heart rate to humans).	- Difficult and expensive to work with. - Ethical considerations.	[17,19,76,77,78]
Small mammals (rodents, mice, rabbits)	-They capture the process of hypoxia-reoxygenation but does not fully model the clinical setting. -Interactions between various cell types.	- Physiologically relevant. -Cheaper compared to big animals. -Easier to genetically manipulate compared to larger animals. -Effective to evaluate therapeutic approaches to regenerate the heart after injury.	-Effectiveness and safety for humans remain to be determined. -Rodent hearts have a much higher intrinsic beating rate, higher cardiac basal metabolism and different electrophysiology compared with the human heart. - Ethical considerations.	[76,79,80]

**Table 2 cells-10-00899-t002:** Major advantages and disadvantages of in vitro models of ischemic heart diseases.

Model	Typical Features	Advantages	Disadvantages	References
2D Cultures (monocellular and multicellular cell layers)	-High control of various confounding factors (temperature, pH, CO_2_). -Widely used to study pathways of IRI and test the candidate therapeutic options.	Monocellular cultures -Testing of the electromechanical properties of individual cardiomyocytes (cardiac physiology). -Individual cardiomyocytes can be controlled by numerous factors such as stress, strain, stiffness. -Effective technique for expanding cell lines Multicellular cultures -Can examine cardiomyocytes culture electrically using microelectrode arrays. -Optimal control over environmental parameters.	Monocellular cultures -Isolated cardiomyocytes can behave differently and show different responses to drugs from cells that are cultured with other cells. -Limited maturity. Multicellular cultures -No cell to cell interaction in 3D and static conditions. - Response to drugs, toxins or signalling modifiers may be misleading.	[8,25,76]
3D Cultures (cardiac spheroids, scaffold-based approaches and organ-on-a-chip models)	-Useful to evaluate more physiologically relevant mechanisms for the prevention and treatment of ischemia/reperfusion injury -Rely on isolated cardiomyocytes from animals, immortalised cell lines, or HiPS-CMs	-Prolonged viability and retain contractile properties. - Mimic key aspects of the phenotypical and cellular heterogeneity as well as microenvironmental aspects. -Cardiac tissue engineering uisng hiPS-CMs aims at promoting cardiac cell maturation and developing a more predictive human tissue model of IRI as well as be patient-specific.	-Expensive cultures. -Tissue culture skills optimal for these cultures are required. -Cell phenotype can be dramatically affected by the culture geometry.	[8,88,89,90,91,92]

**Table 3 cells-10-00899-t003:** Major advantages and disadvantages of in vitro and in vivo models of heart failure (HF).

Model	Typical Features	Advantages	Disadvantages	References
In Vitro 2D Cell Culture	- Monolayer cell cultures of cardiac cells (either transformed cell lines or iPSC-derived cardiac myocytes) can be co-cultured with other cardiac cells to better recapitulate the in vivo cardiac environment. (commonly employed for genetic studies and drug discovery). - Treatment with endothelin-1 is commonly used as a positive control for cardiac hypertrophy (the largest risk factor for heart failure).	- Culturing cells in 2D is significantly cost-effective when using immortallised cell lines. -Both transformed cells and iPSC-derived cells could be human derived. -Extensive literature using transformed cell lines for drug discovery and cardiotoxic effects. - Cardiac myocytes can be employed for studies of genetic mutations in response to hypertension and cardiac hypertrophy.	-Transformed cells have fundamentally altered genomes. - Two-dimensional culturing lacks the full 3D architecture present in vivo (i.e.,interactions with other cells and the ECM). -They cannot fully recapitulate the human heart pathophysiology.	[23,113,117,119,124]
In Vitro *3D* Cell Culture	- Often including a biomaterial (i.e., a hydrogel or biocompatible polymer) for optimal stiffness and electrical signals.	- Improved models of the in vivo physiological, morphological, biochemical and genetic profile. - Engineered 3D environments also use structural features not present in 2D to mimic mechanical cues (i.e., increased afterload).	-Increased complexity of experimental design. -Directly inducing heart failure is still a challenge for in vitro models when compared to in vivo counterparts.	[22,115,116,117]
Small Animal In Vivo	- Transverse aortic constriction (TAC) surgery (greater pressure in the left ventricle and subsequently cardiac hypertrophy, fibrosis as well as cardiac output dysfunction).-In periods of up to 4–6 weeks, this progresses to clinical heart failure.	-TAC procedure is a well established method (it can be easily replicated with consistent results). - Can use transgenic mice-Low maintenance costs when compared to in vivo models in large animals.	-Translatability of results is challenging in small animals. - Features of the heart are functionally different when compared to the human heart. - Slight variations can result in greater pathological stimuli than intended. - Ethical considerations.	[106,107,109,110,111]
Large Animal In Vivo	- A progressive aortic constriction in dogs, sheeps and pigs, is induced in a similar fashion to small animals. - Another method involves tachycardia-induced cardiomyopathy that results in heart failure after several weeks of continuation.	- Increased translatability to human physiology. - Allows live monitoring.	-Research facilities are rarely equipped for significant large animal studies. - Higher costs compared to small animals; - Multidisciplinary teams required for handling. - Ethical considerations.	[18,20,24,106,115,116,125]

## Data Availability

Not applicable.

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
