# Peer review of "Considerations to Model Heart Disease in Women with Preeclampsia and Cardiovascular Disease"

_cells, 2021, doi:10.3390/cells10040899_

Round 1

Reviewer 1 Report

This review summarized different models to study preeclampsia, IR and HF and compared the current in vivo, in vitro and ex vivo models regarding those research. The paper was very well organized and developed. It provides a substantial informative platform for researchers to start the studies of CVD. There are some typos that need to be addressed. other than that,  it is qualified to be accepted. 

Author Response

We would like to thank the Reviewer for the very positive feedback. We hope that the revised manuscript can be found suitable for publication.

Reviewer 2 Report

The authors present a review of the current models employed to study preeclampsia, ischemia reperfusion injury, and heart failure. Overall, this review is well organized and provides a fair assessment of the current state of the field. Because the subject matter is relatively broad, it is not surprising there are some minor holes in the review that could be addressed to better capture some areas that could use a bit more attention. Primarily, the preeclampsia section is missing some details on some important studies within an incredibly broad and diverse literature.

Minor concerns:

  • In the introduction, the authors allude to the concept that preeclampsia may lead to increased rates heart failure in diagnosed women due to IRI, but there does not appear to be any link made later in the text. Either the introduction or discussion should be amended to be clear about the purpose of the review.
  • Lines 41-45, discussion of heterogeneity of preeclampsia phenotypes should include the placental transcriptome analysis by Leavey et al (PMID: 27160201 DOI: 10.1161/HYPERTENSIONAHA.116.07293)
  • Lines 110-116, when discussing rodent models of preeclampsia, while the information discussed is basically true, there are some important caveats, including litter size, gestation time, degree of prenatal development, and hemodynamic issues related to the animals basic physiology when compared to the human. The discussion in the IRI and HF sections are more balanced in this regard, and this section should be brought to that standard
  • Lines 126-133, when discussing the RUPP model, an important drawback to discuss is the fact that largely surgical models such as this are not good models for testing the efficacy of therapeutic intervention, and as a result may be important for recapitulating aspects of the consequences of preeclampsia but not the etiology or whether the condition can be improved
  • Section 2.2.4 is lacking in discussion of endothelial vasodilator production. Good starting points include the following manuscripts:  (Mahdy et al PMCID: PMC2230895 DOI: 10.1111/j.1469-7793.1998.00609.x; Steinert et al PMID: 11923225 DOI: 10.1096/fj.01-0916fje;  Zhang et al PMCID: PMC5505189 DOI: 10.1210/jc.2017-00437

Author Response

Comment 1: The authors present a review of the current models employed to study preeclampsia, ischemia reperfusion injury, and heart failure. Overall, this review is well organized and provides a fair assessment of the current state of the field. Because the subject matter is relatively broad, it is not surprising there are some minor holes in the review that could be addressed to better capture some areas that could use a bit more attention. Primarily, the preeclampsia section is missing some details on some important studies within an incredibly broad and diverse literature.

Response 1: We thank the reviewer for the constructive feedback and we hope we addressed the concerned raised in the updated manuscript.

Comment 2: In the introduction, the authors allude to the concept that preeclampsia may lead to increased rates heart failure in diagnosed women due to IRI, but there does not appear to be any link made later in the text. Either the introduction or discussion should be amended to be clear about the purpose of the review.

Response 2: We thank the reviewer for the comment, and agree that there are no such models that can enable elucidation of the mechanisms linking preeclampsia to heart failure or IRI. However, we have discussed bioinformatics papers that shows a number of overlapping mechanisms between preeclampsia and heart failure or hypertension. We have now updated the Discussion to address the current problem with the lack of models recapitulating this association hence enabling us to study the mechanisms leading to IRI and heart failure post preeclampsia. We have also added a couple of sentences in the introduction with relevant references, which demonstrate increased risk of ischemic heart disease and signs of heart failure with preserved ejection fraction in post-preeclampsia and in preeclampsia models, respectively. We also revised the purpose of the review to discuss the models of these three diseases separately but clearly demonstrating overlapping pathologies.

Changes 2: [lines 638-647] . “Following such an extensive list of models in preeclampsia, IRI and GF, we could not identify a model that fully recapitulates the link between preeclampsia and future CVD in mothers using laboratory models, which is critical to further address the mechanisms leading to these CVDs post-preeclampsia. Studies investigating the increased CVD risk in the offspring born to preeclamptic mothers through fetal reprogramming have been carried out. There are also a lack of model representative of different phenotypes of preeclampsia. This is important for appropriate monitoring and timely treatment in women post preeclampsia who are at risk of developing CVD including IRI and heart failure. A wide range of in vitro, ex vivo and in vivo model systems of individual diseases have helped in discovery and validation of novel therapeutic targets and disease mechanisms. However, there are still large gaps in the knowledge or technology impeding successful translation of the findings from the preclinical into humans’ studies.”

Comment 3: Lines 41-45, discussion of heterogeneity of preeclampsia phenotypes should include the placental transcriptome analysis by Leavey et al (PMID: 27160201 DOI: 10.1161/HYPERTENSIONAHA.116.07293)

Response 3: We have now added the study in the Introduction.

Changes 3: [lines 45-48] “Based on transcriptome analysis, preeclampsia phenotypes have been defined also as: i) “maternal”, ii) “canonical” or iii) “immunologic” (depending on the presence of a healthy placenta and term delivery, typical features of preeclampsia or severe growth restriction and maternal antifetal rejection, respectively).”

Comment 4: Lines 110-116, when discussing rodent models of preeclampsia, while the information discussed is basically true, there are some important caveats, including litter size, gestation time, degree of prenatal development, and hemodynamic issues related to the animals basic physiology when compared to the human. The discussion in the IRI and HF sections are more balanced in this regard, and this section should be brought to that standard

Response 4: We appreciate this comment from the reviewer. We have expanded this section to discuss main differences in physiology between animal and human models. We have already discussed inter-species differences in trophoblast invasion and placental morphology in different sections.

Changes 4: [lines 107-116] “There are substantial physiological differences between animal models of preeclampsia and human preeclampsia in terms of gestational period (rats ~22 days/mice 19-20 days versus 40 weeks in humans), litter size (6-8 in mice; 8-18 in rats and often single in humans), degree of prenatal development, as well as immune, metabolic and hemodynamic profiles. Nevertheless, rodents display similar cardiovascular adaptations to pregnancy including increase in glomerular filtration rate, vasodilation and cardiac output, and reduction in systemic vascular resistance; rodents are the most frequently used for developing in vivo preeclampsia models. The findings identified in relation to preeclampsia (i.e. targets, treatments or biomarkers) in small animal models need to be validated in nonhuman primate models of preeclampsia before progressing to humans.”

Comment 5: Lines 126-133, when discussing the RUPP model, an important drawback to discuss is the fact that largely surgical models such as this are not good models for testing the efficacy of therapeutic intervention, and as a result may be important for recapitulating aspects of the consequences of preeclampsia but not the etiology or whether the condition can be improved

Response 5: We agree with the reviewer and have added a sentence to this effect.

Change 5: [Line 145-147] “As this model of preeclampsia is surgically induced it does not facilitate elucidation of the mechanisms of etiology of preeclampsia and testing of preventative treatments.”

Comment 6: Section 2.2.4 is lacking in discussion of endothelial vasodilator production. Good starting points include the following manuscripts:  (Mahdy et al PMCID: PMC2230895 DOI: 10.1111/j.1469-7793.1998.00609.x; Steinert et al PMID: 11923225 DOI: 10.1096/fj.01-0916fje;  Zhang et al PMCID: PMC5505189 DOI: 10.1210/jc.2017-00437

Response 6: We thank the reviewer for the suggestion. Whilst we discussed briefly the release of ET-1 vasoconstrictor in preeclampsia we have now expanded to include other reduced vasodilation mechanisms observed in the presence of preeclampsia. For this, we used the most recent review on vascular mechanisms in preeclampsia (PMID 32762557).

Changes 6: [Lines 312-315]  “These circulating inflammatory and oxidative stress factors lead to vascular damage and reduction in endothelium-derived vasodilators including nitric oxide or prostacyclin and increase in vasoconstrictors, endothelin 1 (ET-1) and thromboxane A2 . Increase in vascular cell adhesion molecule-1 (VCAM-1) is also observed in preeclampsia.”

Reviewer 3 Report

Dear Author

I really acknowledge the interesting review you want to publish.  It is an extensive review of the in vivo and in vitro models for a future study of PE and cardiovascular disease. This is a relevant issue and deeper research needs to be done on that field.

Few comments need to be done.

I consider the tables should be more friendly, probably less text should be considered.

Moreover, I wonder whether the effect of the fetal compartment should be mentioned and developed. 

I encourage you to refine these items. Your article should be then ready to be reanalysed.

Author Response

We thank the reviewer for the very positive feedback.

Comment 1: the tables should be more friendly.

Response 1: all table have been updated and we hope the reviewer finds them improved as well.

Comment 2: the effect of the fetal compartment.

Response 2. We thank the Reviewer again for the comment. We have now added an paragraph in the Discussion regarding this.

Changes 2: [lines 833-837] "While the etiology of preeclampsia is still not fully uncovered, there is evidence that the fetal environment plays a major role (PMID: 26104643). Among the plethora of potential biomarkers identified in previous studies, future studies will be critical for the identification of better markers for improved classification and potential prevention and therapeutic approaches (doi:10.4172/0974-276X.S10-001)."